# Development of a Multi-Channel Wearable Heart Sound Visualization System

**DOI:** 10.3390/jpm12122011

**Published:** 2022-12-04

**Authors:** Binbin Guo, Hong Tang, Shufeng Xia, Miao Wang, Yating Hu, Zehang Zhao

**Affiliations:** 1School of Biomedical Engineering, Dalian University of Technology, Dalian 116024, China; 2Liaoning Key Lab of Integrated Circuit and Biomedical Electronic System, Dalian University of Technology, Dalian 116024, China; 3School of Information and Communication Engineering, Dalian University of Technology, Dalian 116024, China

**Keywords:** phonocardiogram, acoustic sensor, acoustic mapping, sensor arrays, multichannel measurement system

## Abstract

A multi-channel wearable heart sound visualization system based on novel heart sound sensors for imaging cardiac acoustic maps was developed and designed. The cardiac acoustic map could be used to detect cardiac vibration and heart sound propagation. The visualization system acquired 72 heart sound signals and one ECG signal simultaneously using 72 heart sound sensors placed on the chest surface and one ECG analog front end. The novel heart sound sensors had the advantages of high signal quality, small size, and high sensitivity. Butterworth filtering and wavelet transform were used to reduce noise in the signals. The cardiac acoustic map was obtained based on the cubic spline interpolation of the heart sound signals. The results showed the heart sound signals on the chest surface could be detected and visualized by this system. The variations of heart sounds were clearly displayed. This study provided a way to select optimal position for auscultation of heart sounds. The visualization system could provide a technology for investigating the propagation of heart sound in the thoracic cavity.

## 1. Introduction

The heart is one of the most complex organs in the human body. Cardiac mechanical activity is caused by ECG activity, including the contraction and diastole of the heart, which causes blood to flow inside the heart and engender heart sounds. The origins of heart sounds include the vibrations caused by blood impacting on the heart valves and walls, and blood turbulence vibrations. Previous studies proved that the vibrations of the heart valves correspond to the components of heart sounds [1]. Abnormal valves cause significant changes in heart sound characteristics. The first heart sound (S1) is generated in systole, which is mainly related to the closure of mitral and tricuspid valves. The second heart sound (S2) is produced in the late stage of left ventricular contraction, which is mainly associated with the closure of aortic and pulmonary valves. The probability of the third heart sound (S3) in healthy people is very low, while the fourth heart sound (S4) is generally morbid [2]. The above heart sounds are spread to the chest surface through body tissues such as lungs, bones, muscles, and fat. Differences in signal sources and propagation paths contribute to the significant characteristic variations in the heart sound signals acquired from different regions.

Since Laennec invented the first stethoscope based on the acoustic characteristics of the pipeline in 1816, the acquisition technology had been constantly updated [3]. Traditional stethoscopes were difficult to cope with the interference of ambient noise, and the accuracy of diagnosis was influenced by the experience and ability of the physician [4]. An electronic stethoscope converts heart sounds to digital signals via AD converters, and the signals are processed with algorithms to help physicians in clinical diagnosis. Heart sound sensors are commonly based on microphones, piezoelectric sensors and acceleration sensors. With the development of technology, intelligent sensors have functions such as adaptive noise reduction, intelligent signal analysis, and wireless transmission [5]. 

In recent years, researchers have paid more attention to the visualization of heart sounds. Multi-channel heart sound signals are characterized by high spatial resolution and high sampling rate. The signals provide much more information than a single channel signal by employing big data analysis, which enables research on the spatial distribution of heart sound sources and the propagation model of heart sound signals. Research has been conducted since the last century, and as early as 1980 Chihara et al. used 25 sensors to acquire multi-channel heart sound signals and used an imaging method for heart diagnosis [6]. Later, Okada drew chest wall maps of heart sounds and murmurs to show their distribution patterns over time [7]. With the development of signal acquisition technology, Leong et al. designed a system for a real-time cardiac acoustic map based on a DSP board [8]. Kawamura et al. acquired heart sounds of the chest wall simultaneously with using 64 small accelerometers and estimated the propagation path of signals on the chest wall using the delay time of adjacent acquisition points [9]. Nogata et al. employed 63 acceleration sensors and one ECG sensor to acquire heart sound signals and an ECG signal, and analyzed them from different frequency bands with 24 h monitoring [10]. In recent years, Sapsanis et al. developed a synchronous multi-channel wearable system to create a cardiac acoustic map that could generate spatiotemporal images [11]. To explore the relationship between heart sound vibration and valves, Munck used 16 triaxial sensors to analyze vibration waves based on the sagittal axis, a single cardiac cycle, and four reference points [12]. To improve the quality of heart sound signals, Pasha et al. applied beamforming and channel equalization technology to multi-channel recordings of heart sounds. This method effectively improved the SNR (signal-to-noise ratio) of the signals [13]. In 2021, a new wireless multichannel stethoscope system was developed by Zhang et al. for easier acquisition of multichannel heart sound signals and the ECG signal. This system served to monitor cardiovascular and cardiopulmonary diseases [14]. 

The above research has focused on acquiring multichannel signals, but precise locations of the sensors placed on the chest surface were not sufficiently considered in the mapping. Some studies used oversized sensors, resulting in low spatial resolution [12,14]. The distribution of sensors in some studies was less rational, and the signal analysis was limited. In this paper, image registration was used to locate the positions of the sensors on the chest surface accurately during the acquisition process. Cubic spline interpolation was applied to draw the cardiac acoustic map to improve its accuracy. In addition, the quality of the maps was related to the sensors, such as frequency response, size, position, and quantity of sensors. In this paper, a novel microphone heart sound sensor based on MEMS technology was developed with small size, high sensitivity, and high signal quality. The sensor enabled effective acquisition of heart sounds at different locations on the chest surface. Furthermore, the sensor had adjustable gains and initial filtering to the heart sound signal.

In this paper, a heart sound visualization system was successfully developed and presented in the form of cardiac acoustic map. A wearable multi-channel heart sound acquisition device based on a novel sensor was used in the system. Image registration, filtering, and cubic spline interpolation were used to draw the map. Compared to previous studies, the proposed system can adapt to variations of sensor location when the wearable devices are applied to different subjects. Our method has high accuracy in studying the transmission mechanism of heart sounds, which help physicians to understand the principles of heart sound propagation and select optimal position for auscultation of heart sounds.

## 2. Materials and Methods

### 2.1. System Architecture

The visualization system architecture is illustrated in Figure 1. The novel heart sound sensor contained a microphone, a linear amplifier, and filter circuits. The wearable multichannel signal acquisition device consisted of 72 heart sound sensors and one ECG analog front end (AFE). The sensors were embedded in a wearable vest with an inflatable airbag. The sensors could be pressed against skin by the airbag with equal air pressure. Eight digital signal acquisition cards (USB2881, ART Technology, China) were used to simultaneously sample heart sound signals and the ECG signal with 16-bit A/D resolution. The sampling rate was up to 250 kHz. 

Signal preprocessing techniques, band-pass filtering and wavelet noise reduction, were used to tune the signals. An image of a human skeleton was used as the background for the cardiac acoustic map. The positions of each sensor on the chest surface during data acquisition were accurately marked and then transformed to the corresponding positions in the human skeleton image. Finally, the pre-processed signals and the position information were used to draw the cardiac acoustic maps based on cubic spline interpolation. 

### 2.2. Sensor and Wearable Vest

The novel heart sound sensor converted mechanical vibration into an electrical signal. It consisted of three electronic components, i.e., an ICS-40300 microphone, a high-pass filter, and a MAX9814 linear amplifier. The main parameters of these components are shown in Table 1. 

As shown in Figure 2a, the ICS-40300 is a MEMS microphone for sensing heart sound. In Figure 2b, the MAX9814 is a high-quality amplifier, which is featured as a three-stage amplifier. The first stage is a low noise amplifier (LNA) with a fixed 12 dB gain. The second stage is a variable gain amplifier (VGA) that is controlled by automatic gain control (AGC) and can adjust the gain to 0 dB or 20 dB automatically. To avoid signal distortion, the AGC was turned off to keep a fixed 20 dB gain of the VGA. The third stage is an output amplifier, which has three selections of gain (8, 18 and 28 dB). It is programmed through a single trilevel logic input. In this work, the gain of the third stage was set at 18 dB. The overall gain of the MAX9814 was 50 dB. 

The schematic of the heart sound sensor is shown in Figure 3. The capacitor C_1_ was a 0.1 μF unit used to decouple the noise from the power supply. The signal was transmitted via capacitor C_in_ into the MAX9814. The C_in_ was a 0.082 μF DC-blocking capacitor, which formed a high-pass filter with the input impedance $R_{M}$ of the MAX9814. The cut-off frequency $f_{C}$ of the high-pass filter is given by: (1)$$f_{C}=1/(2\pi \times C_{\mathrm{in}}\times R_{M})$$
where $R_{M}$ is 100 kΩ, and the result of $f_{C}$ is 19.4 Hz. As mentioned above, the LNA had a 12 dB gain. The pins of TH and MICB were connected to disable the AGC function, so the gain of VGA became 20 dB. The gain of the output amplifier was selected as 18 dB by connecting the pin of GAIN and GND. A 0.82 nF AC-coupling capacitor C_out_ was connected to the pin of MICOUT to eliminate the DC offset. The input impedance R_L_ of the data acquisition card (USB2881, ART Technology, Beijing, China) and C_out_ formed a high-pass filter. The cut-off frequency $f_{O}$ of the filter is given by:(2)$$f_{O}=1/(2\pi \times C_{\mathrm{out}}\times R_{L})$$
where R_L_ is 10 MΩ, and the result of $f_{O}$ is 19.4 Hz. The pins of VDD and BIAS bypassed to GND with a 1 μF capacitor and a 0.47 μF capacitor, respectively. The pin of CG connected a 2.2 μF capacitor with GND to ensure zero offset at the output. The pins of VDD and $\bar{\mathrm{SHDN}}$ were connected to power VDD. All remaining pins were connected to GND.

The printed circuit board (PCB) of the sensor is shown in Figure 4a. It had a round design with a 10 mm diameter and three external connections: VDD, GND and Output. Figure 4c shows the packaged sensor. The sensor package included two parts. First, the PCB was embedded in a flexible rubber housing with a diameter of 18 mm and a height of 7 mm, as shown in Figure 4b. The housing had a cylindrical cavity at the bottom that was used for coupling heart sounds. Second, the PCB was sealed with hot melt adhesive to reduce loss of energy. The size of the PCB and sensor are shown in Figure 4e. The maximum diameter was less than 20 mm, and its weight was about 3 g. The ECG signal was used as a reference signal to distinguish the cardiac cycles. The ECG AFE was produced by the SICHIRAY company (see Figure 4d). Figure 4f shows the wearable vest. The vest has 72-channel heart sound sensors. The sensors were within the red square, and the rest of the sensors are not studied in this paper. One channel of ECG AFE was placed on its inner side. The air pressure in the airbag was set to 1.8 kPa to press the sensors against skin. Figure 5 shows the multichannel wearable heart sound visualization system, including a desktop computer, data acquisition cards, a vest with multi-channel heart sound sensors and ECG AFE, a power supply system, and an air pump for airbag inflation.

### 2.3. Sensor Test

The sensor performance test included sensor consistency, frequency response, and SNR. The test was performed in a quiet room of 15 m^2^. The DC power supplied to the sensor was set to 3.3 V. Two heart sound sensors (Sensor A and Sensor B) were randomly selected and placed side by side 5 cm away from a loudspeaker. The loudspeaker was used to play sinusoidal signals; frequency j started from 20 Hz and increased in step of 5 Hz to 200 Hz. During the test, the loudness of the loudspeaker remained unchanged. The time duration of signal acquisition was 40 s and the sampling rate was 2 kHz. The power spectral densities (PSDs) of the acquired signals were calculated and their center frequencies were estimated from spectrum analysis. 

The sensor consistency was measured by the difference rate $\delta_{V,j}$ in amplitude and the difference rate $\delta_{f,j}$ in frequency between the output signals. They were defined as
(3)$$\delta_{V,j}=\frac{\left|V_{A,j}-V_{B,j}\right|}{V_{B,j}}$$
(4)$$\delta_{f,j}=\frac{\left|f_{A,j}-f_{B,j}\right|}{f_{B,j}}$$
where $V_{A,j}$ and $V_{B,j}$ are the amplitudes of output signals of Sensor A and Sensor B when the sinusoidal signal frequency is j. $f_{A,j}$ and $f_{B,j}$ are the frequencies of Sensor A and Sensor B when the sinusoidal signal frequency is j.

The SNR is defined as
(5)$$SNR=10\mathrm{lg}(\frac{P_{1}}{P_{2}})$$
where $P_{1}$ is the signal power and $P_{2}$ is the noise power.

SNRs were compared between the novel heart sound sensor and a sensor provided by ADInstruments. The subject involved in SNR comparison was a 28-year-old male, 82 kg in weight and 182 cm in height. As shown in Figure 6, during the test, the subject lay on his back and was quiet. Figure 6 shows the application of the two sensors.

### 2.4. Signal Preprocessing

Multichannel heart sound signals were synchronously sampled and the sampling rate $f_{s}$ was selected as 10 kHz. The ECG signal was defined as $s_{\mathrm{ECG}}(n)$ and the heart sound signal of channel i was named $s_{i}(n)$.

Firstly, the heart sound signals were mean normalized to remove baseline drift.
(6a)$$D_{\max}=\max(s_{i}(n)),\;i=1:72,\;n=1:N$$
(6b)$$D_{\min}=\min(s_{i}(n)),\;i=1:72,\;n=1:N$$
(7)$$s_{i,D}(n)=\frac{s_{i}(n)-m_{i}}{D_{\max}-D_{\min}}$$
where $D_{\max}$ is the maximum value, $D_{\min}$ is the minimum value, N is the number of samples, and $m_{i}$ is the mean of $s_{i}(n)$.

According to prior knowledge, the dominant frequencies of normal heart sounds are from 20 Hz to 200 Hz [15]. We used a 4-order Butterworth filter with a passband of 20–200 Hz to filter the signals. To avoid the phase deviation introduced by the filter, a bidirectional filter with zero phase distortion was used. Signals were filtered in the forward direction, then the filtered signals were reversed and ran through the filter again.

Wavelet transform has high frequency resolution and low time resolution in the low frequency part, and opposite properties in the high frequency part [16]. Multiscale analysis of the signals by scaling and translation of the wavelet principal function provided an excellent portrayal of the non-smooth characteristics of the signals. Based on this technique, the noise could be separated from the heart sound signals [17]. In the process of decomposing the signals at a sampling rate of 10 kHz, the frequency band of the detailed levels (1 to 8) were approximately 2500–5000, 1250–2500, 625–1250, 312.5–625, 156.25–312.5, 78.13–156.25, 39.07–78.13, and 19.5–39.07 Hz. The seventh and eighth detailed levels contained most of the energy of the signals. Thus, the level of the wavelet was selected as 7, and the coif-5 wavelet was adopted as the mother wavelet because of its good analytical performance for the signals [16,17,18]. The fixed threshold selection rule Sqtwolog and soft threshold were used in the MATLAB wavelet toolbox to process the signals [19]. The threshold selection rule ‘Sqtwolog’ is a fixed threshold equal to the square root of two times the logarithm of the length of the signals. For the soft threshold, the elements whose absolute values are lower than the threshold are set to zero, and then the nonzero coefficients are shrunk towards zero. Although hard thresholding is the simplest method, soft thresholding can produce better results than hard thresholding [16].

The ECG signal consisted of a low amplitude signal superimposed on high common voltage and noise. The frequency bandwidth of the ECG signal was 0.05–100 Hz, and the amplitude range was 0–3.0 mV. We focused on R-wave peak detection, and a frequency bandwidth of 0.5–40 Hz was chosen. The ECG signal was down sampled to 200 Hz and subjected to noise reduction by a zero-phase Butterworth bandpass filter. Then it was up-sampled to 10 kHz. The ECG R-waves were detected by the Pan Tompkins algorithm and used as time references to identify cardiac cycles. 

### 2.5. Channel Location

During signal acquisition, the sensors left impressions on the chest surface due to the pressure of the airbag. The subject’s chest was photographed and the impressions marked with green discs (Figure 7a), named sensor markers. Specific locations on the human skeleton were selected as reference points for image registration, named sensor registration markers. As shown in Figure 7a, those marked with red discs included the sternoclavicular joint $\left(p_{1},q_{1}\right)$, the tenth rib of the right thorax $\left(p_{2},q_{2}\right)$, the sixth rib of the right thorax $\left(p_{3},q_{3}\right)$ and the sixth rib of the left thorax $\left(p_{4},q_{4}\right)$. Similarly in Figure 7d, the same locations in the human skeleton image were marked with green discs and named skeleton registration markers, including $\left(v_{1},k_{1}\right)$, $\left(v_{2},k_{2}\right)$, $\left(v_{3},k_{3}\right)$, and $\left(v_{4},k_{4}\right)$. 

Figure 7a was processed by the HSV (hue, saturation, value) color space to extract the sensor registration markers (Figure 7b) and the sensor markers (Figure 7c). The same technique was used to extract the skeleton registration markers (Figure 7e) from the skeleton image (Figure 7d). The coordinates of the markers in the image were represented by the centers of the circles, which were detected based on the Circle Hough Transform algorithm.

Rigid transformations were employed to realign the chest image (Figure 7a) and the skeleton image (Figure 7d), including rotational, scaling, and translational transformations. Rotational transformation was used for image angle registration. The coordinates of the sensor markers channel i were defined as $\left(w_{i},h_{i}\right)$. The new coordinates after rotation transformation were defined as $\left(w_{i,\alpha },h_{i,\alpha }\right)$.
(8)$$\theta_{1}=\cot((q_{4}-q_{3})/(p_{4}-p_{3}))$$
(9)$$\theta_{2}=\cot((k_{4}-k_{3})/(v_{4}-v_{3}))$$
(10)$$w_{i,\alpha }=\frac{w_{i}}{\cos(\theta_{1})}\times \cos(\theta_{2})$$
(11)$$h_{i,\alpha }=\frac{h_{i}}{\sin(\theta_{1})}\times \sin(\theta_{2})$$
where $\theta_{1}$ is the angle between the direction of the line connecting sensor registration markers $\left(p_{3},q_{3}\right)$ and $\left(p_{4},q_{4}\right)$ with the horizontal direction. $\theta_{2}$ is the angle between the direction of the line connecting sensor registration markers $\left(v_{3},k_{3}\right)$ and $\left(v_{4},k_{4}\right)$ with the horizontal direction. Similarly, the coordinates of sensor registration markers were taken into Equations (10) and (11) to obtain the markers $\left(p_{1,\alpha },q_{1,\alpha }\right)$, $\left(p_{2,\alpha },q_{2,\alpha }\right)$, $\left(p_{3,\alpha },q_{3,\alpha }\right)$, and $\left(p_{4,\alpha },q_{4,\alpha }\right)$ after rotational transformation.

Scaling transformation was used for image size registration. After rotational transformation, the skeleton image was scaled to obtain new coordinates $\left(w_{i,\beta },h_{i,\beta }\right)$.
(12)$$\eta_{x}=(v_{4}-v_{3})/(p_{4,\alpha }-p_{3,\alpha })$$
(13)$$\eta_{y}=(k_{2}-k_{1})/(q_{2,\alpha }-q_{1,\alpha })$$
(14)$$w_{i,\beta }=\eta_{x}\times w_{i,\alpha }$$
(15)$$h_{i,\beta }=\eta_{y}\times h_{i,\alpha }$$
where $\eta_{x}$ is the variation of the abscissa, and $\eta_{y}$ is the variation of the ordinate. Similarly, the coordinates of sensor registration markers after rotational transformation were taken into Equations (14) and (15) to obtain the markers $\left(p_{1,\beta },q_{1,\beta }\right)$, $\left(p_{2,\beta },q_{2,\beta }\right)$, $\left(p_{3,\beta },q_{3,\beta }\right)$, and $\left(p_{4,\beta },q_{4,\beta }\right)$ after scaling transformation.

Translational transformation was used for the matching and positioning of the sensor markers and human skeleton image. The last coordinates $\left(w_{i,\gamma },h_{i,\gamma }\right)$ were obtained as follows:(16)$$\varphi_{x}=v_{3}-p_{3,\beta }$$
(17)$$\varphi_{y}=k_{1}-q_{1,\beta }$$
(18)$$w_{i,\gamma }=w_{i,\beta }+\varphi_{x}$$
(19)$$h_{i,\gamma }=h_{i,\beta }+\varphi_{y}$$
where $\varphi_{x}$ was the pixel value translated horizontally, and $\varphi_{y}$ was the pixel value translated in the vertical direction.

As shown in Figure 7f, the red circles show the positions in the skeleton image after registration. Finally, the coordinates of the sensors in the chest image were mapped to the skeleton image, as shown in Figure 7g.

The visualization system resulted in high accuracy of the cardiac acoustic maps by accurately locating the positions of the sensors. Subjects were analyzed individually in the study, considering obesity, skeleton structure, and other factors. The registration operation enabled the system to adapt to people with different shapes. Whether a person is tall, short, fat or thin, the system can display his/her cardiac acoustic map on the standard skeleton image.

### 2.6. Cardiac Acoustic Mapping

Cardiac acoustic maps were drawn using pre-processed signals and coordinates in the skeleton image. Cubic spline interpolation of the cardiac acoustic maps was performed according to the signal’s modulus, with various amplitudes corresponding to different colors. High-modulus areas were bright and low-modulus areas were shaded. The modulus values corresponded to blue, green, yellow, and red in order from low to high. 

The modulus $e_{i}(n)$ of the heart sound signal of channel i at instant n was
(20)$$e_{i}(n)=\left|s_{i,D}(n)\right|$$

By combining the coordinates $\left(w_{i,\gamma },h_{i,\gamma }\right)$ with the modulus $e_{i}(n)$, a three-dimensional data $Z(w_{i,\gamma },h_{i,\gamma },e_{i}(n))$ was constructed.

Based on $Z(w_{i,\gamma },h_{i,\gamma },e_{i}(n))$, cubic spline interpolation was performed to obtain the interpolated three-dimensional data G(W,H,E(n)), where E(n) is the modulus of the pixel (W,H) at time n.

The color dial based on HSL (hue, saturation, lightness) color space was set, and ε(n) was the hue of the pixels at index n. The value of ε(n) was 240 to 0, saturation was set to 1.0, and lightness was set to 50%. ε(n) varied inversely with the modulus.
(21)$$\varepsilon_{g}(n)=E(n)/E_{m}$$
(22)$$\varepsilon (n)=240\times (1-\varepsilon_{g}(n))$$
where $E_{m}$ is the maximum value of modulus, $\varepsilon_{g}(n)$ is the normalized modulus of pixel (W,H) at index n, and ε(n) is the hue after conversion of pixel (W,H) at index n. Finally, the HSL color space was converted into RGB space to draw the cardiac acoustic maps.

## 3. Results and Discussions

### 3.1. Heart Sound Sensor Consistency

Figure 8 shows the test results for sensor A and sensor B. Figure 8a indicates the frequency response in the range 20 to 200 Hz of Sensor A. The results show that the signals output from the sensor in this frequency range were stable. As shown in Figure 8b, the difference rates in amplitude of the two sensors at the same frequency were about 0.01. The frequency differences of the sensors are illustrated in Figure 8c, from which it can be found that the sensor output signals had almost the same frequency. Figure 8d shows the difference rates of the frequency between the output signals, and the ideal signals were about 0.001.

The novel heart sound sensor developed in our study was compared with the heart sound sensor produced by ADInstruments, and a 30 Hz high-pass filter was used for initial filtering of the signals during signal acquisition. 

Figure 9a shows the heart sound signal in the third intercostal space of the left chest acquired by the novel sensor. The power of the heart sound signal was 588.6, and the power of the noise between S1 and S2 was 3. According to Equation (5), the SNR of the heart sound signal obtained by the novel sensor in this acquisition area was 22.9 dB. 

Figure 9b shows the heart sound signal acquired by the ADInstruments sensor. The power of the signal was 22.6 and the power of the noise was 0.16. The SNR of the heart sound signal acquired by the sensor in the region was about 21.5 dB.

Table 2 shows the features of the novel heart sound sensor. In summary, the heart sound sensor designed in our study had excellent performance. The sensors had good consistency with each other, and when applied to the chest surface could acquire the original heart sound signal in the area. 

### 3.2. Multichannel Heart Sound Signals

The visualization system acquired 72 heart sound signals from the chest surface of a 27-year-old male of 183 cm in height and 65 kg in weight. The signals are shown in Figure 10, and the numbers below image are channel indexes. Significant differences among channels can be seen. Generally speaking, the signal amplitude was stronger in the region closer to the heart, and the signal intensity was reduced with positions away from the heart.

### 3.3. Cardiac Acoustic Map 

As shown in Figure 11, the heart sound signal and ECG signal were acquired using the visualization system. The blue curve in Figure 11 is the ECG signal, from which P, Q, R, S, and T waves could be distinguished. The red curve in Figure 11 is the heart sound signal of channel 22 in Figure 10, from which S1 and S2 could be distinguished. As a typical example, the system showed the cardiac acoustic maps for some stages of the first and the second heart sound, as seen in Figure 12 and Figure 13. The four stages of S1 are indicated as “S1-1”, “S1-2”, “S1-3”, and “S1-4”. The four stages of S2 are expressed as “S2-1”, “S2-2”, “S2-3”, and “S2-4”.

The cardiac acoustic maps of S1-1 in Figure 12 corresponded to the timing of the signal from 0.204 s to 0.228 s, as seen in Figure 11, and the sampling interval of the maps was 0.002 s. The stage of S1-1 occurred early in S1, after the R wave of the ECG signal in the time domain. At this time point, the ventricle entered systole, and the atrioventricular valve began to close [20,21], and the modulus of the heart sound signal gradually increased. As shown in Figure 12, the areas of heart sound modulus concentration were from the third to fourth intercostal spaces of the left chest. The second and third intercostal spaces on the right side of the sternum also had significant high-modulus heart sound signals, but were weaker compared with the left chest.

The cardiac acoustic maps of S1-2 in Figure 12 corresponded to the signals of 0.228 s to 0.254 s in Figure 11. The S1-2 occurred in the middle of the S1, after the S wave of the ECG signal. At this time, the depolarization of the left and right ventricles was completed. The ventricles entered isovolumic systole, and the mitral and tricuspid valves were closed [20,21]. As observed in Figure 12, the heart sound signals were stronger in modulus and higher in frequency compared to the S1-1 stage. The areas of heart sound modulus concentration were from the third to fourth intercostal spaces of the left chest, as well as the second and third intercostal spaces of the right chest. The brightest area of the heart sound modulus gradually moved down to the apex of the heart.

The cardiac acoustic maps of S1-3 in Figure 12 corresponded to the signals of 0.254 s to 0.280 s in Figure 11. The S1-3 period was still in the isovolumic systolic of the ventricle, and both the mitral and tricuspid valves were closed [20,21,22]. The modulus of the signals was the same as that of the S1-2 period, but the frequency was higher. The areas of concentration of heart sound modulus in the left chest were between the third and fifth intercostal spaces.

The cardiac acoustic maps of S1-4 in Figure 12 corresponded to the signal starting from 0.282 s till 0.322 s in Figure 11. The stage of S1-4 occurred at the end of the S1 stage, and the intraventricular pressure reached its highest. At this time, the aortic and pulmonary valves gradually opened. The heart began to enter the ejection period, and the heart sound modulus gradually decreased [20,21]. The highest regions of heart sound modulus were mainly concentrated on the third to fifth intercostal spaces of the left chest, and gradually reached the end of the S1 stage. The period lasted longer in S1.

Figure 12 demonstrates the characteristics of the modulus of the heart sounds on the chest surface over time for each period. The modulus of the S1 signal was mainly concentrated on the third to fifth intercostal spaces in the left chest, as well as the second and third intercostal spaces in the right chest. The brightest regions of the heart sound modulus in that period shifted down from the third intercostal space in the left chest to the apex of the heart with time.

The cardiac acoustic maps of the S2 of the subject ae shown in Figure 13. This divides S2 into four stages, S2-1, S2-2, S2-3, and S2-4. The duration of S2 was shorter, and the sampling interval between images was 0.001 s.

The cardiac acoustic maps of S2-1 in Figure 13 correspond to the signals from 0.469 s to 0.482 s in Figure 11. The S2-1 period occurred in early S2, when the ventricles entered diastole, the blood flow decelerated abruptly in the aorta and pulmonary artery impinged on the valves to produce S2. At this time, the aortic and pulmonary valves began to close [22,23,24]. As shown in Figure 13, the modulus of the signals gradually increased during S2-1, and the higher modulus areas were concentrated on the third and fourth intercostal spaces of the left chest, and the second intercostal space at the right edge of the sternum.

The cardiac acoustic maps of S2-2 in Figure 13 correspond to the signals in Figure 11 from 0.482 s to 0.502 s. The S2-2 period was closely connected to the S2-1 period. In this period the aortic and pulmonary valves closed abruptly due to the impact of blood flow, and the heart sounds had high frequency and modulus [23,24]. From Figure 13, the modulus of heart sounds during S2-2 gradually increased to the highest and gradually decreased to lowers. The high-modulus areas in this period initially appeared in the third and fourth intercostal spaces in the left chest. Subsequently, the high-modulus areas were concentrated on the second to fifth intercostal spaces of the left chest. The signal in this period had the longest duration and the highest modulus and frequency in S2.

The cardiac acoustic maps in Figure 13 for S2-3 correspond to the signals in Figure 11 from 0.502 s to 0.508 s. The S2-3 period was about to enter the end of isovolumic diastole, when the intraventricular pressure gradually decreased, and the ventricle was about to enter the filling period [25]. As can be seen in Figure 13, the heart sound signals during S2-3 became progressively weaker in modulus, and the distribution of higher modulus heart sound signals was wider in the regions. These regions were in the third, fourth and fifth intercostal spaces of the left chest.

The cardiac acoustic maps of S2-4 in Figure 13 correspond to signals from 0.509 s to 0.536 s in Figure 11. The S2-4 period occurred in the end of S2, when the intraventricular pressure gradually reached its minimum and was about to enter the filling period. As shown in Figure 13, the signal modulus was overall weaker during S2-4, and the higher modulus areas were concentrated on the third and fourth intercostal spaces of the left chest [26].

Compared with previous related research, the wearable multi-channel heart sound acquisition device designed by us had a larger acquisition area covered chest surface and more heart sound sensors. The trend of changes of heart sound modulus on the chest surface over time could be clearly observed from the images. A movie built by the time series of images could be obtained in the future to show time varying vibration of heart sounds. This technique may enable physicians to discover abnormal vibration occurring in the heart, such as valve disease, hypertension, and abnormal vascular reconstruction.

## 4. Conclusions

The current work showed the propagation relationship of heart sound modulus on the chest surface using cardiac acoustic maps. A novel heart sound sensor with high SNR was designed and embedded into a multi-channel wearable heart sound signal acquisition device, which could be used to acquire multi-channel heart sound and ECG signals simultaneously. The visualization system improved the spatial resolution of human heart sounds and provided a technical basis for the study of heart sound propagation and clinical diagnosis. The signal processing part of the system contained noise reduction, channel location, and cardiac acoustic mapping.

The cardiac acoustic maps contained the temporal, spatial, and modulus characteristics of the heart sound signals. From the Figure 11, Figure 12 and Figure 13, the modulus variation characteristics on the chest surface at different time intervals can be observed. Figure 12 shows the modulus changes of the S1, while Figure 13 focuses on the modulus changes of the S2. From the changes in the maps, there were obvious differences in the modulus and the propagation area of the signals in different periods, and the differences were closely related to the motion of the heart. In clinical practice, based on cardiac acoustic maps, doctors could find the strongest areas of modulus at each moment of the heart sound cycle. This could help physicians locate the best auscultation area and make better diagnoses. This may allow the physician to make a diagnosis of the subject’s heart based on the cardiac acoustic maps. This system has potential applications. It may allow production of an acoustic movie and enable physicians to disclose abnormal vibration occurring in the heart at an early stage, such as valve disease, hypertension, and abnormal vascular reconstruction.

## Figures and Tables

**Figure 1 jpm-12-02011-f001:**
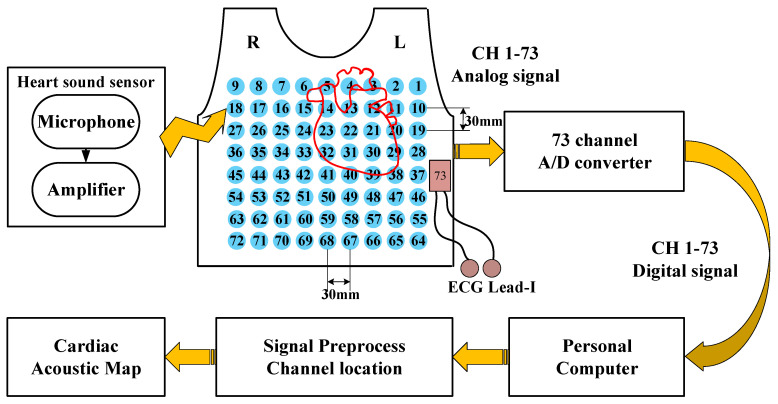
Architecture of the visualization system.

**Figure 2 jpm-12-02011-f002:**
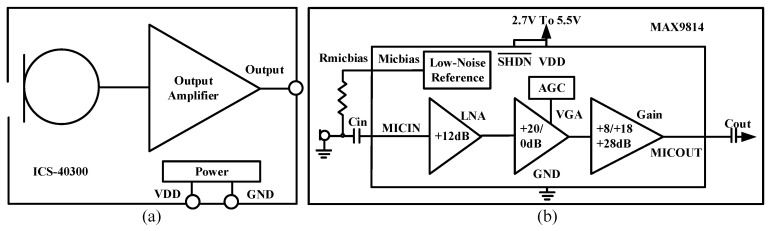
Schematics of (**a**) ICS-40300 and (**b**) MAX9814.

**Figure 3 jpm-12-02011-f003:**
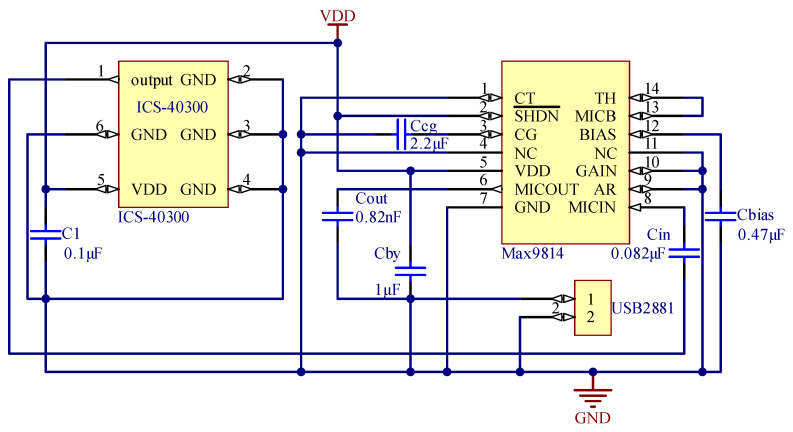
Schematic of the heart sound sensor.

**Figure 4 jpm-12-02011-f004:**
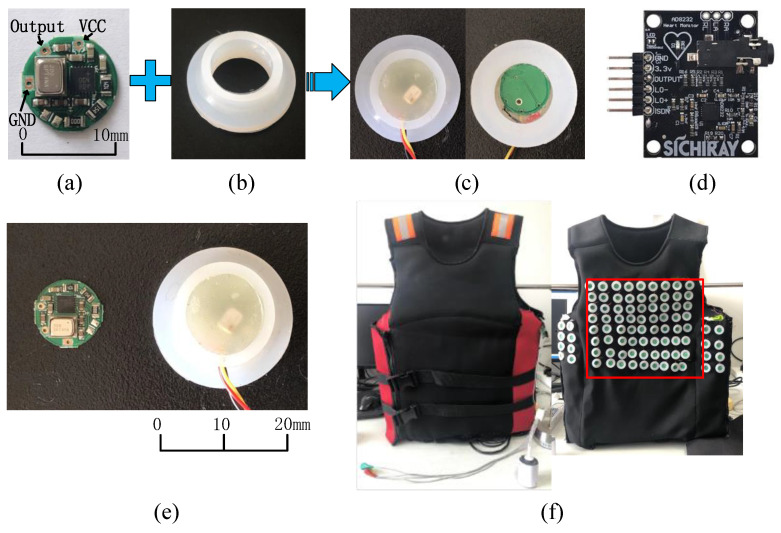
Multi-channel wearable heart sound acquisition device. (**a**) Sensor PCB, (**b**) rubber housing, (**c**) packaged sensor, (**d**) ECG AFE, (**e**) size of the sensor and the PCB, (**f**) vest and the sensors arrangement on the inside of the vest.

**Figure 5 jpm-12-02011-f005:**
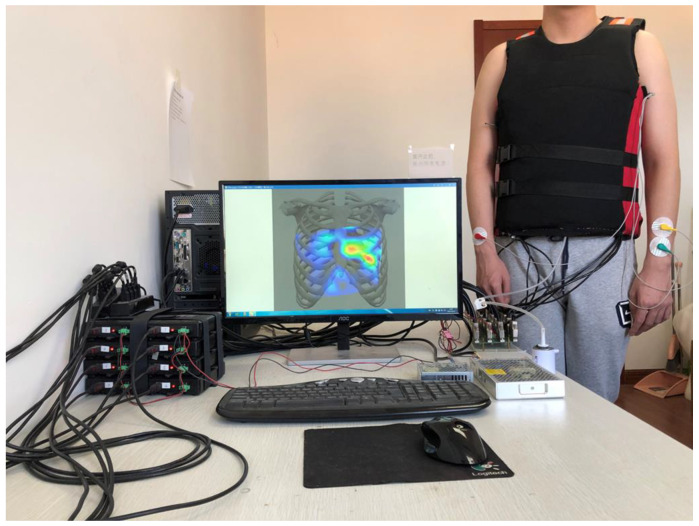
Multi-channel wearable heart sound visualization system.

**Figure 6 jpm-12-02011-f006:**
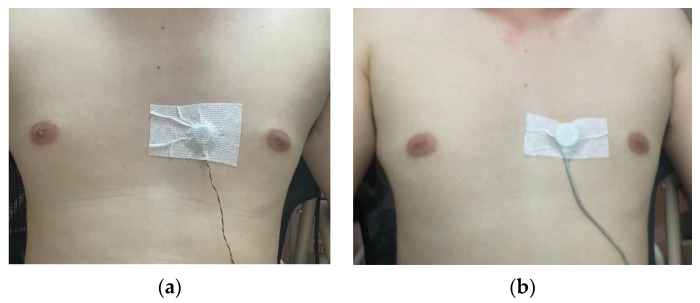
Sensors tested on the chest surface. (**a**) The novel heart sound sensor; (**b**) the sensor from ADInstruments.

**Figure 7 jpm-12-02011-f007:**
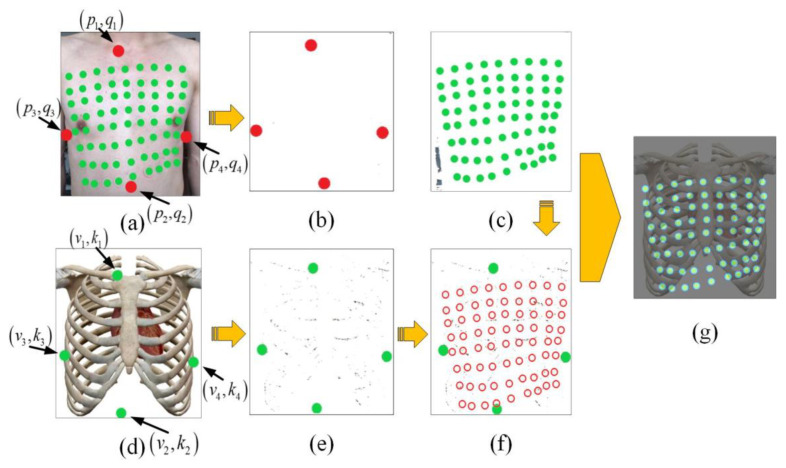
Diagram of channel location. (**a**) Image of marked chest surface; (**b**) sensor registration markers extracted from the chest image; (**c**) sensor markers extracted from the chest image; (**d**) image of human skeleton with the skeleton registration markers; (**e**) skeleton registration markers extracted from the human skeleton image; (**f**) sensor arrangement after registration between sensor registration markers and skeleton registration markers; (**g**) image of the sensor arrangement in the skeleton image after registration.

**Figure 8 jpm-12-02011-f008:**
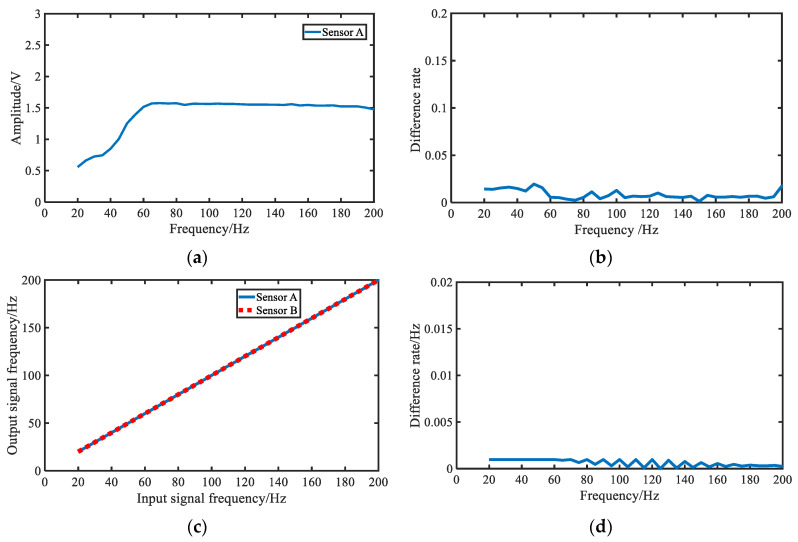
Results of the sensor tests. (**a**) Frequency response in test range of Sensor A; (**b**) difference rates in amplitude of the output signals between the test sensors at the same frequency; (**c**) differences in output signals frequency between the test sensors; (**d**) difference rates of frequency between the output signals of the test sensor and the ideal signals.

**Figure 9 jpm-12-02011-f009:**
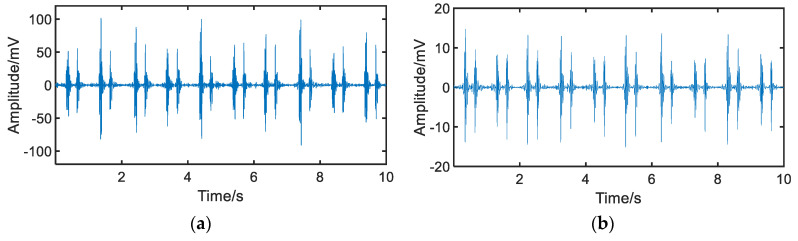
Heart sound signals acquired by the two sensors from the same area on the chest surface. (**a**) heart sound signal acquired by the novel sensor; (**b**) heart sound signal acquired by the ADInstruments sensor.

**Figure 10 jpm-12-02011-f010:**
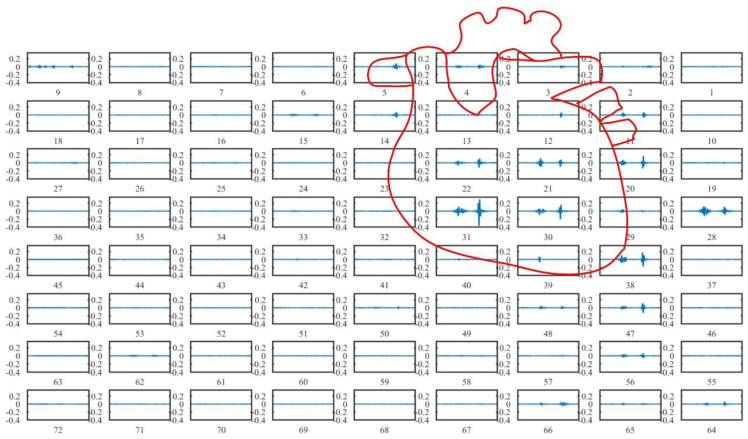
72-channel heart sound signals at the chest surface of a 27-year-old healthy male of height 183 cm and weight 65 kg.

**Figure 11 jpm-12-02011-f011:**
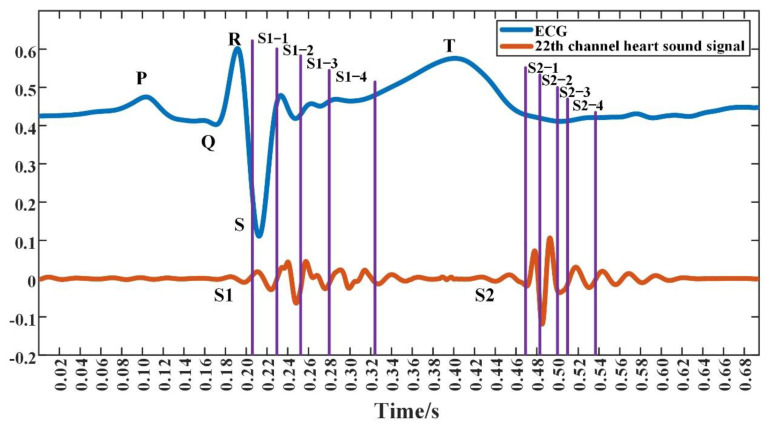
ECG signal and heart sound signal acquired on channel 22 on a male aged 27, 183 cm in height and 65 kg in weight. The first heart sound (S1) and the second heart sound (S2) are each divided into four stages.

**Figure 12 jpm-12-02011-f012:**
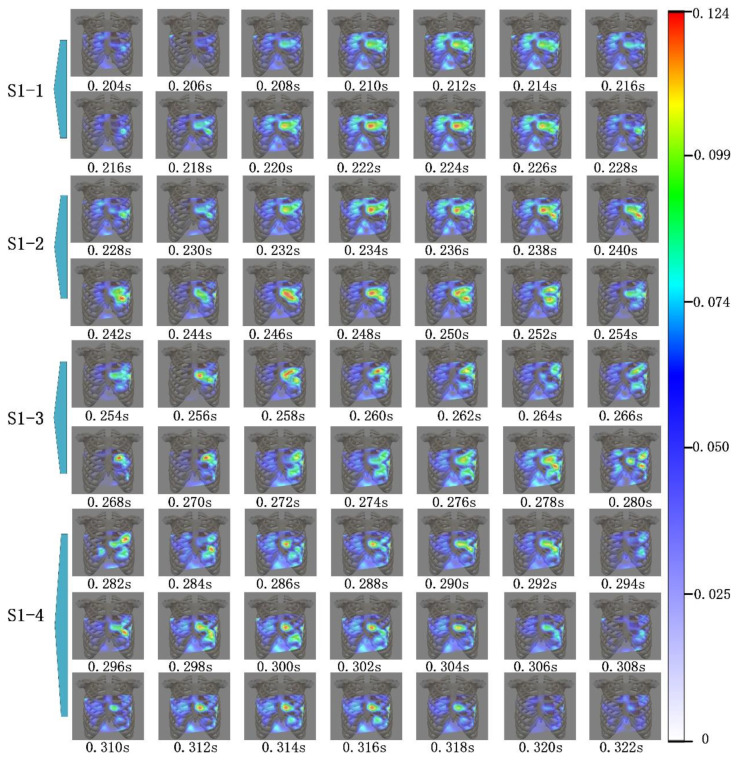
Cardiac acoustic maps of S1. The stages S1-1, S1-2, S1-3, and S1-4 are indicated in Figure 11.

**Figure 13 jpm-12-02011-f013:**
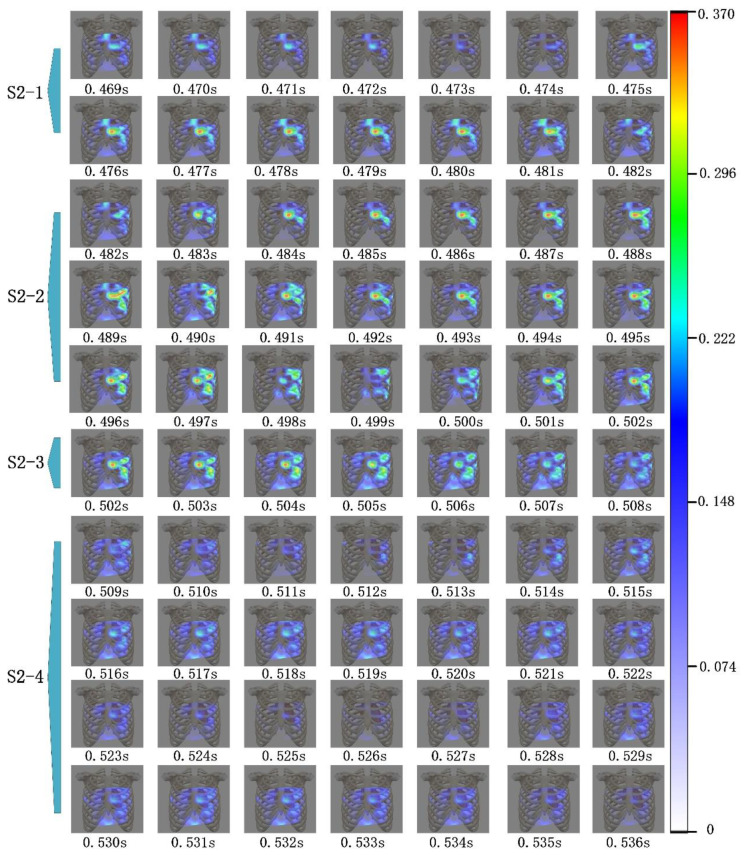
Cardiac acoustic maps of S2. The stages of S2-1, S2-2, S2-3, and S2-4 are indicated in Figure 11.

**Table 1 jpm-12-02011-t001:** Parameters of electronic components.

Parameters	ICS-40300	MAX9814
Size (mm)	4.72 × 3.76 × 3.5	3 × 3
Frequency response (Hz)	6–20,000	20–20,000
DC power supply (V)	1.5–3.63	2.7–5.5
SNR (dB)	63	64
Sensitivity (dBV) Input impedance (kΩ) Out impedance (Ω)	−45 −200	−100 50

**Table 2 jpm-12-02011-t002:** Features of the novel heart sound sensor.

Parameters	The Novel Heart Sound Sensor
Size (mm)	16 × 16 × 7
Frequency response (Hz)	20–200
DC power supply (V)	3.3–3.63
SNR (dB)	22.9
Out impedance (Ω)	50

## Data Availability

Data available in a publicly accessible repository that does not issue DOIs. Publicly available datasets were analyzed in this study. This data can be found here: https://github.com/BinBinGuo-Signal/Cardiac-acoustic-map, accessed on 20 October 2022.

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
