# Peer review of "Development of a Multi-Channel Wearable Heart Sound Visualization System"

_jpm, 2022, doi:10.3390/jpm12122011_

Round 1

Reviewer 1 Report

The manuscript is interesting and as far as my knowledge on the subject reaches, it contributes novelties to the state of the art in the field of knowledge. However, before being published, some minor revision should be addressed.

1.- Some English sentences are difficult to understand, so I suggest that the wording (Syntax and semantics) of the manuscript be reviewed in greater detail.

2.- line 33 paragraph INTRODUCTION. Put a reference that supports the sentence "Previous studies proved that the vibrations of the heart valves correspond to the components of heart sounds []."

3.- Line 44 paragraph INTRODUCTION. Put a reference that supports the sentence "Since Laennec invented the first stethoscope based on the acoustic characteristics of the pipeline in 1816, the acquisition technology had been constantly updated []."

4.- Line 82 paragraph INTRODUCTION. Put a reference that supports the sentence "Some studies used oversized sensors, resulting in low spatial resolution []."

5.- Line 83 paragraph INTRODUCTION. Put a reference that supports the sentence "The distribution of sensors in some studies was less rational and the signal analysis was limited []."

6.- Line 237 paragraph 2.4 Signal Preprocessing. Put a reference that supports the sentence "Wavelet transform has high frequency resolution and low time resolution in the low frequency part and the opposite properties in the high frequency part []."

7.- Line 241 paragraph 2.4 Signal Preprocessing. Put a reference that supports the sentence "Based on this technique, the noise could be separated from the heart sound signals []."

8.- Line 246-47 paragraph 2.4 Signal Preprocessing.  When the authors point out "... its good analytical performance...". Explain what the authors did to decide that there was a good analytical performance of the signals."

9.- Line 248 paragraph 2.4 Signal Preprocessing.  Explain in more detail (explicitly) the sentence "The fixed threshold and soft threshold were used to process."

Author Response

Thank you very much for carefully reviewing our article. Your suggestions and comments are very meaningful to us. After careful consideration and study, we make the following replies to your comments.

Point 1: Some English sentences are difficult to understand, so I suggest that the wording (Syntax and semantics) of the manuscript be reviewed in greater detail.

 Response 1: We carefully checked the content of the text, corrected any syntax and semantics errors in the text, and corrected and redrawn any problems in the figures.

Point 2: line 33 paragraph INTRODUCTION. Put a reference that supports the sentence "Previous studies proved that the vibrations of the heart valves correspond to the components of heart sounds []."

Response 2: We put the article [1] Arslan, O., et al., Automated detection of heart valve disorders with time-frequency and deep features on PCG signals. Biomedical Signal Processing and Control, 2022. 78(103929) as the reference to supports the sentence "Previous studies proved that the vibrations of the heart valves correspond to the components of heart sounds."

Point 3: Line 44 paragraph INTRODUCTION. Put a reference that supports the sentence "Since Laennec invented the first stethoscope based on the acoustic characteristics of the pipeline in 1816, the acquisition technology had been constantly updated []."

 Response 3: We put the article [3]. Takashina, T., et al., New Stethoscope with Extensible Diaphragm. Circulation Journal, 2016. 80(9): 2047-2049 as the reference to supports the sentence "Since Laennec invented the first stethoscope based on the acoustic characteristics of the pipeline in 1816, the acquisition technology had been constantly updated."

 Point 4: Line 82 paragraph INTRODUCTION. Put a reference that supports the sentence "Some studies used oversized sensors, resulting in low spatial resolution []."

 Response 4: We put the articles [12]. Munck, K., et al., Multichannel seismocardiography: an imaging modality for investigating heart vibrations. Physiological Measurement, 2020. 41(11500111) and Zhang, X., et al., Development of a novel wireless multi-channel stethograph system for monitoring cardiovascular and cardiopulmonary diseases. IEEE Access, 2021. 9: 128951-128964 as the references to support the sentence "Some studies used oversized sensors, resulting in low spatial resolution."

Point 5: Line 83 paragraph INTRODUCTION. Put a reference that supports the sentence "The distribution of sensors in some studies was less rational and the signal analysis was limited []."

 Response 5: We put the article [11]. Sapsanis, C., et al., StethoVest: A simultaneous multichannel wearable system for cardiac acoustic mapping. 2018 IEEE Biomedical circuits and systems conference (BIOCAS): Advanced Systems for Enhancing Human Health, 2018: 191-194 as the reference to supports the sentence "The distribution of sensors in some studies was less rational and the signal analysis was limited."

Point 6: Line 237 paragraph 2.4 Signal Preprocessing. Put a reference that supports the sentence "Wavelet transform has high frequency resolution and low time resolution in the low frequency part and the opposite properties in the high frequency part []."

 Response 6: We put the article [16]. Messer, S.R., et al., Optimal Wavelet Denoising for Phonocardiograms. Microelectronics Journal, 2001. 32(12): 931-941 as the reference to supports the sentence "Wavelet transform has high frequency resolution and low time resolution in the low frequency part and the opposite properties in the high frequency part."

Point 7: Line 241 paragraph 2.4 Signal Preprocessing. Put a reference that supports the sentence "Based on this technique, the noise could be separated from the heart sound signals []."

 Response 7: We put the article [17]. Mondal, A., et al., A Noise Reduction Technique Based on Nonlinear Kernel Function for Heart Sound Analysis. IEEE Journal of Biomedical and Health Informatics, 2018. 22(3): 775-784 as the reference to supports the sentence "Based on this technique, the noise could be separated from the heart sound signals."

Point 8: Line 246-47 paragraph 2.4 Signal Preprocessing. When the authors point out "... its good analytical performance...". Explain what the authors did to decide that there was a good analytical performance of the signals."

Response 8: In this paper, we learned that coif-5 wavelet have good performance for processing heart sound signals based on previous studies. So we put articles [16]. Messer, S.R., et al., Optimal Wavelet Denoising for Phonocardiograms. Microelectronics Journal, 2001. 32(12): 931-941 and [17]. Mondal, A., et al., A Noise Reduction Technique Based on Nonlinear Kernel Function for Heart Sound Analysis. IEEE Journal of Biomedical and Health Informatics, 2018. 22(3): 775-784 as references to support the sentence "... its good analytical performance...". The coif5 wavelet was used as the mother wavelet in both of the above-mentioned papers and the superiority of this wavelet was illustrated.

Point 9: Line 248 paragraph 2.4 Signal Preprocessing. Explain in more detail (explicitly) the sentence "The fixed threshold and soft threshold were used to process."

Response 9: In this paper, the fixed threshold and soft threshold are the threshold parameters of wavelet. According to the article [16]. Messer, S.R., et al., Optimal Wavelet Denoising for Phonocardiograms. Microelectronics Journal, 2001. 32(12): 931-941. The threshold selection rule ‘Sqtwolog’ is a fixed form threshold equal to the square root of two times the logarithm of the length of the signal. The soft threshold is a common method of thresholding signal. For the soft thresholding ,the elements whose absolute values are lower than the threshold are set to zero, and then the nonzero coefficients are shrunk towards zero. Although hard thresholding is the simplest method, soft thresholding can produce better results than hard thresholding.

Reviewer 2 Report

In the manuscript, the authors designed a multi-channel wearable heart sound visualization system based on novel heart sound sensors for imaging cardiac acoustic map was. This research may have potential applications in investigating the propagation of heart sound in the thoracic cavity. The paper could be published after minor revision addressing the following points:

1. In section 2.3, in the sensor test, the author started the test at 60Hz. However, the heart sound signals have a frequency range of 20-600Hz, so it is not reasonable to start testing at 60Hz.

2. In section 2.4, the authors claim that the frequency of normal heart sounds is between 30 and 150Hz. Looking up reference 13, I could not find the corresponding argument. In the process of heart sound research, it is generally believed that the heart sound of normal people is 20-200Hz, and that of patients with heart disease is 20-600Hz. Therefore, the author's use of 30-150Hz bandpass filter is unreasonable.

3. In Section 3.1, the author starts with a test frequency of 60Hz. But the test diagram shown in Figure 8 looks more like it started at 20Hz.

4. In Section 3.1, the authors state in Table 2 that the novel heart sound sensor has a frequency response of 20-2000Hz and a sensitivity of -45dB. The author's frequency response test did not reach 20-20000Hz, and did not do sensitivity test. So the author can't make that claim.

5. In section 3.3, the heart sound waveform shown in Figure 11 is an abnormal heart sound waveform. In normal heart sounds, the amplitude of the first heart sound is larger than that of the second heart sound. The heart sound waveform shown in Figure 9 is a normal one.

6. The references in the article are too old.

Author Response

Thank you very much for carefully reviewing our article. Your suggestions and comments are very meaningful to us. After careful consideration and study, we make the following replies to your comments.

Point 1: In section 2.3, in the sensor test, the author started the test at 60Hz. However, the heart sound signals have a frequency range of 20-600Hz, so it is not reasonable to start testing at 60Hz.

Response 1: Thanks for your comments, the frequency range was changed to 20-200 Hz in the sensor test, which is within the range of normal human heart sound signals.

Point 2: In section 2.4, the authors claim that the frequency of normal heart sounds is between 30 and 150Hz. Looking up reference 13, I could not find the corresponding argument. In the process of heart sound research, it is generally believed that the heart sound of normal people is 20-200Hz, and that of patients with heart disease is 20-600Hz. Therefore, the author's use of 30-150Hz bandpass filter is unreasonable.

Response 2: In this paper, the study subject is normal people, and according to the comments of the reviewer, we adjusted the filtering range to 20-200Hz after finding the relevant papers.Mean while , We found that the spectrogram in the reference [15]. Kumar, D., et al., Noise Detection During Heart Sound Recording. 2009 Annual International Conference of the IEEE Engineering in Medicine and Biology Society, 2009: 3119-3123 shows that the main frequency range of heart sounds in normal people is 20-200 Hz.

Point 3: In Section 3.1, the author starts with a test frequency of 60Hz. But the test diagram shown in Figure 8 looks more like it started at 20Hz.

Response 3: In Figure 8, we have added a scale of horizontal coordinates to indicate the frequency range of the test signal.

Point 4: In Section 3.1, the authors state in Table 2 that the novel heart sound sensor has a frequency response of 20-2000Hz and a sensitivity of -45dB. The author's frequency response test did not reach 20-20000Hz, and did not do sensitivity test. So the author can't make that claim.

Response 4: The data for the sensitivity of the sensors used in this paper was obtained from the data sheets of the ICS-40300, and the relevant tests were not performed due to the lack of relevant test instruments. According to the reviewer's comment, the data of sensor sensitivity is deleted from this paper. At the same time, the frequency response range of the sensor is adjusted to 20-200Hz.

Point 5: In section 3.3, the heart sound waveform shown in Figure 11 is an abnormal heart sound waveform. In normal heart sounds, the amplitude of the first heart sound is larger than that of the second heart sound. The heart sound waveform shown in Figure 9 is a normal one.

Response 5: The heart sounds in Figure 9 and Figure 11 in the text belong to different normal subjects, respectively. The S1 and S2 of the heart sound signals are closely related to the spatial distribution of the heart, and the S1 and S2 in different regions can be significantly different. In addition, the S1 and S2 of the heart sound signal may be affected by the propagation process in different subjects because of the fat thickness and individual differences, thus showing different S1 and S2. In addition, the S1 and S2 of the heart sound signal will be different in different acquisition areas of the same subject, as can be seen in Figure 10 in the article, the heart sound signal in channel 21 has a significant difference. There is still no definitive conclusion that S1 is certainly larger than S2 in normal people heart sounds.

Point 6: The references in the article are too old.

Response 6: We replace references [15]. Castelli, E., et al., Heart sounds in the polygraphic evaluation of the phases of the cardiac cycle. Critical review. II. The 1st sound. Archivio di patologia e clinica medica, 1967. 44: 310-30, [16]. Delman, A., et al., Hemodynamic Correlates of Cardiovascular Sounds. Annual Review of Medicine, 1967. 18(1): 139-158, [17]. Mconald, D.A., Hemodynamics. Annual Review of Physiology, 1968. 30(1): 525-556, [18]. Bonavita, C., et al., Heart sounds in the polygraphic evaluation of the phases of the cardiac cycle. Critical review. 3. The 2d sound. Archivio di patologia e clinica medica, 1967. 44: 392-404 in the original article to references [20].            Polat, A., The comprehensive analysis of the determination of wavelet function-level pair for the decomposition and reconstruction of artificial S1 heart signals by using multi-resolution analysis. Biomedical Signal Processing and Control, 2021. 70(103055), [21]. Nath, M., et al., Detection and localization of S-1 and S-2 heart sounds by 3rd order normalized average Shannon energy envelope algorithm. Proceedings of the Institution of Mechanical Engineers Part H-Journal of Engineering in Medicine, 2021. 235(09544119219981086): 615-624, [22]. Jimenez-Gonzalez, A., Timing the opening and closing of the aortic valve by using the phonocardiogram envelope: a performance test for systolic time intervals measuring. Physiological measurement, 2021. 42(025004), [23]. Altuve, M., et al., Fundamental heart sounds analysis using improved complete ensemble EMD with adaptive noise. Biocybernetics and Biomedical Engineering, 2020. 40(1): 426-439, [24]. Saeidi, A., F. Almasganj and M. Shojaeifard, Automatic cardiac phase detection of mitral and aortic valves stenosis and regurgitation via localization of active valves. Biomedical Signal Processing and Control, 2017. 36: 11-19. The newly added articles are the latest studies in recent years.

Reviewer 3 Report

very good work, may be very useful for anticipate the diagnosis of hypertension, valvulopaties, restrictive cardiomyopaties. The system could be improved by including the apical area up to the mid or posterior axillary line in the acoustic signal detection area. The graphic representation of the patient's left lateral decubitus heart sound, or right in the case of  dextrocardia could also be sampled.

Author Response

Thank you very much for your suggestions and comments. In our future research work, we will continue to explore the cardiac acoustic maps of human heart sounds in different regions. This research can be applied to clinical applications in the future to help doctors to process and analyze the cardiac acoustic maps. That canbe used to diagnose and predict heart diseases through artificial intelligence methods. Meanwhile, there are still many unsolved mysteries about the propagation of heart sounds, and the results of this paper will provide a strong technical basis for it.